# Protective Efficacy of Multiple Epitope-Based Vaccine against *Hyalomma anatolicum*, Vector of *Theileria annulata* and Crimean–Congo Hemorrhagic Fever Virus

**DOI:** 10.3390/vaccines11040881

**Published:** 2023-04-21

**Authors:** Abhijit Nandi, Vandana Solanki, Vishvanath Tiwari, Basavaraj Sajjanar, Muthu Sankar, Mohini Saini, Sameer Shrivastava, S. K. Bhure, Srikant Ghosh

**Affiliations:** 1Division of Parasitology, ICAR-Indian Veterinary Research Institute, Izatnagar 243122, India; drabhijitnandi@gmail.com (A.N.); nishrohi008@gmail.com (M.); drsankarm@gmail.com (M.S.); 2Department of Biochemistry, Central University of Rajasthan, Bandarsindri, Ajmer 305817, India; vtphd03@gmail.com (V.S.); vishvanath@curaj.ac.in (V.T.); 3Veterinary Biotechnology Division, ICAR-Indian Veterinary Research Institute, Izatnagar 243122, India; bksvet@gmail.com (B.S.); sameer_vet@rediffmail.com (S.S.); 4Division of Biochemistry, ICAR-Indian Veterinary Research Institute, Izatnagar 243122, India; mohini@ivri.res.in (M.S.); sdbhure@gmail.com (S.K.B.)

**Keywords:** *Hyalomma anatolicum*, in silico, multi-epitope peptide, peptide vaccine, protective efficacy

## Abstract

*Hyalomma anatolicum* is the principal vector for *Theileria annulata*, *T. equi*, and *T. Lestoquardi* in animals and the Crimean–Congo hemorrhagic fever virus in humans. Due to the gradual loss of efficacy of the available acaricides against field tick populations, the development of phytoacaricides and vaccines has been considered the two most critical components of the integrated tick management strategies. In the present study, in order to induce both cellular and humoral immune responses in the host against *H. anatolicum*, two multi-epitopic peptides (MEPs), i.e., VT1 and VT2, were designed. The immune-stimulating potential of the constructs was determined by in silicoinvestigation on allergenicity (non-allergen, antigenic (0.46 and 1.0046)), physicochemical properties (instability index 27.18 and 35.46), as well as the interaction of constructs with TLRs by docking and molecular dynamics analysis. The immunization efficacy of the MEPs mixed with 8% Montanide^TM^ gel 01 PR against *H. anatolicum* larvae was determined as 93.3% and 96.9% in VT1- and VT2-immunized rabbits, respectively. Against adults, the efficacy was 89.9% and 86.4% in VT1- and VT2-immunized rabbits, respectively. A significant (*p* < 0.001) reduction in the anti-inflammatory cytokine (IL-4) and significantly higher IgG response was observed in a VT1-immunized group of rabbits as compared with the response observed in the control group. However, in the case of the VT2-immunized rabbits, an elevated anti-VT2 IgG and pro-inflammatory cytokine (IL-2) (>30 fold) along with a decreased level of anti-inflammatory cytokine IL-4 (0.75 times) was noted. The efficacy of MEP and its potential immune stimulatory responses indicate that it might be useful for tick management.

## 1. Introduction

Ticks are obligatory blood-feeding ectoparasites of animals and humans. They are the most potent vector of animal diseases caused by protozoans, spirochetes, bacteria, rickettsia, and viruses [1] than any other arthropods due to their multiple-host-feeding pattern and lengthy feeding periods [2]. The global annual economic losses due to tick infestations have been determined as USD 22–30 billion [3]. Recently, the annual economic impact of ticks and tick-borne diseases (TTBDs) in India is estimated to be approximately USD 787.63 million [4]. Although there are many tick species prevailing in India, *Rhipicephalus microplus* and *Hyalomma anatolicum* are the two most economically significant species infesting Indian livestock [5]. Among them, *H. anatolicum* is the principal vector of *Theileria annulata*, *T. equi*, and *T. Lestoquardi* [5,6]. It also acts as a vector of the Crimean–Congo hemorrhagic fever virus (CCHFV) [7,8,9]. The conventional practice of tick management is mainly focused on treating animals with commercially available chemical acaricides. Yet, this method has provided a limited degree of success with several pitfalls, such as acaricidal residues in meat and milk, a selection of drug insensitive ticks, and has adverse impacts on the environment due to chemical residues [10,11]. The potentiality of immunoprophylactic methods against ticks has already been proven to reduce the use of acaricides [12,13,14]. Ghosh et al. [15] opined that in the present alarming scenario of the establishment of multi-acaricide-resistant ticks in India, there is a need to develop effective phytoacaricides and immunization protocol against ticks as an alternative sustainable strategy.

After the successful introduction of two vaccines for the control of *R. microplus* in field conditions, the efficacy of the vaccines against different strains of the tick species was evaluated with variable results [16,17,18,19]. The differential efficacy and no or limited cross-protective efficacy were the two most important attributes of a low level of adoption of the vaccine technology for field use [20,21,22].

Upon reviewing the literature, it appears that the selection of specific tick protein antigens (epitopes) is a usefuland advantageous strategy for vaccine production. The peptides targeting epitopes which augments precise immune responses to the essential immunogenic segment of a protein antigen can be designed [23]. Malonis et al. [24] advocated that a multi-epitope peptide vaccine may hold the key to future parasite vaccine development.

Previously, three synthetic peptides (SBm4912, SBm7462, and SBm19733) derived from Bm86 of *R. microplus* were used to immunize cattle and an efficacy of >81% against challenge infestations was reported using SBm7462 peptide [25]. A synthetic 20 amino acid peptide from the ribosomal protein P0 was around 90% effective against *R. microplus* and *R. sanguineus* challenge infestations on the bovine and canine host, respectively [26,27]. Further, Rodríguez-Mallon et al. [28] reported around 90% efficacy when dogs and cattle were vaccinated with pP0–Bm86 and challenged with *R. sanguineus* and *R. microplus*, respectively.

In the present study, an attempt was madeto design multi-epitope peptide constructs of immunodominant peptides by immunoinformatic approach. The peptides were preferentially picked based on previous laboratory data [29,30,31] and its efficacy was evaluated against experimental challenge infestations of *H. anatolicum* larvae and adults.

## 2. Materials and Methods

### 2.1. Experimental Animals

Twenty-eight young NewZealand white (NZW) rabbits (animal experiments permission no. IAEC/26.07.2021/S5) were raised in clean rabbit-rearing steel cages (25 × 20 inches) by providing ad libitum feed (multi grains, vegetables, grass) and fresh water. In addition, cross-bred (*Bos indicus* × *B. taurus*) male calves were maintained in atick-proof cattle shed withinthe Division of Parasitology of the institute as per the standard guidelines.

### 2.2. Maintenance of H. anatolicum Ticks for Challenge Study

The acaricide-susceptible and *Theileria annulata*-free *H. anatolicum* (IVRI-2 strain) was maintained by the ear bag protocols of Ghosh and Azhahianambi [32]. NZW rabbits and cross-bred male calves were used for maintaining the tick stages.

### 2.3. In Silico Designing of Multi-Epitopic Peptide Constructs

Based on our previous findings, the proteins were selected and amino acid sequences of ferritin-2 (FER2) and tropomyosin (TPM) were retrieved from the NCBI database (ALJ92580.1 and AMB19058.1) [30]. The tick vitellogenin receptor (VgR) (NOV41407 Porto Alegre strain, Brazil) was identified as major vaccine target due to its involvement in the maturation of ovary and transmission of pathogens [31]. Hence, VgR along with FER2 and TPM were selected for vaccine design.

#### 2.3.1. Prediction of Nature and Localization

To ascertain the occurrence and cleavage site locus of the peptides, the proteins were characterized using SignalP 6.0 server (http://www.cbs.dtu.dk/services/SignalP/ (accessed on 15 January 2020)) [33]. Subsequently, the protein sequences were processed by DeepLoc1.0 (http://www.cbs.dtu.dk/services/DeepLoc/ (accessed on 15 January 2020)) to predict localization [34].

#### 2.3.2. Prediction of BandT-Cell Epitopes and Construction of Multi-Epitope Peptide Sequence

BepiPred-2.0 and ABCpred were used to predict the B-cell epitopes of varying lengths [35,36].

The cytotoxic T lymphocyte (CTL) and helper T-cell (HTL) epitopes were predicted using a specific server. Nine residual lengths of CTL epitopes were deliberately selected using NetMHCpan 4.1 (allele of BoLA) [37]. While 15-mer HTL epitopes are ascribable to NetMHCIIpan 4.1 server (HLA class II based) [37]. The IFNγ epitopes were predicted from all the short-listed epitopes using IFNepitope server (http://crdd.osdd.net/raghava/ifnepitope/ (accessed on 17 January 2020)) [38].

Two MEP (multi-epitopic peptide) sequences were constructed using conserved, exposed potent epitopes. The MEPs were constructed by combining different epitopes using GPGPG linkers. To make the constructs more functional, PRIDE (universal T cell epitope), CPP (cell-penetrating peptide), and EAAAK linker at N and C terminals were also added (Figure 1).

#### 2.3.3. Evaluation of Antigenic and Physicochemical Properties of the Constructs

TheVaxiJenV2.0 (http://www.ddg-pharmfac.net/vaxijen/VaxiJen/VaxiJen.html (accessed on 20 February 2020)) and AllerTOP v2.0 tools were used to predict the antigenicity and allergenicity of the proteins [39]. The physicochemical parameters such as amino acid composition, theoretical pI, in vivo half-life, instability index, molecular weight, grand average of hydropathicity (GRAVY), and aliphatic index were assessed by employing the web server ProtParam (http://web.expasy.org/protparam/ (accessed on 20 February 2020)) [40].

#### 2.3.4. Prediction of Secondary and Tertiary Structure of the Constructs

The input sequence was processed by the RaptorX server, which enables anticipation of 2D and 3D protein structures coupled with solvent accessibility (ACC) and disordered regions (DISO). Moreover, the server provides insight into the quality of constituted 3D models by elaborating different confidence score indices, viz. *p*-value, uGDT (un-normalized GDT), and GDT (global distance test), and root mean square deviation (RMSD) [41].

#### 2.3.5. Refinement and Validation of the Tertiary Structure

The 3D model of the multi-epitopic peptides were later refined by GalaxyRefine server (http://galaxy.seoklab.org/cgi-bin/submit.cgi?type=REFINE (accessed on 22 February 2020)) [42]. After selecting a refined 3D structure model, the tertiary structure was validated for the identification of potential errors. ProSA-web, SAS (sequence annotated by structure, https://www.ebi.ac.uk/thornton-srv/databases/sas/ (accessed on 22 February 2020)) [43], SWISS-MODEL structure assessment tools, and MolProbity server were also used for validation [44].

#### 2.3.6. Molecular Docking of the Constructs with Bovine TLR-2, TLR-4, and TLR-9

The CPORT server was used for the prediction of active sites in toll-like receptors (TLRs) and vaccine constructs [45]. The prediction of the molecular interactions/probable immunogenic trigger, structural complex-based docking of designed multi-epitopic constructs peptide with TLRs, were performed using Haddock 2.4. It generated a table of the 10 best refined structural models for the given task. Later, the best model was selected based on the lowest inter-molecular energies.

#### 2.3.7. Molecular Dynamics Simulation (MDS)

The molecular dynamics simulation study was conducted for the MEPs–TLR complex by recruiting the iMODSwebserver (http://imods.Chaconlab.org/ (accessed on 1 April 2020)). The server processed the assigned job by NMA (normal mode analysis) approach in dihedral coordinates (torsional space) for setting biological macromolecules in naturally reproduced collective motion. Another (molecular dynamics simulation) MDS study was also carried out to obtain information on construct stability using the MDWEB server (https://mmb.irbbarcelona.org/MDWeb//index.php (accessed on 1 April 2020)).

### 2.4. Synthesis, Formulation and Storage of MEPs

Two sets of MEPs were custom synthesized and formulated on a lysine-based backbone with N-terminal acetylation and C-terminal amidation. The individual peptide was chemically ligated to the lysine core. Masses of the synthesized peptides were checked, and finally, post-amidation and linear forms of the two peptides were synthesized with >96% purity (LifeTein, New Jersey, NJ, USA). The molecular weight of the VT1 and VT2 was estimated as 7997.09 Da and 6893.98 Da, respectively (data provided by LifeTein, New Jersey, NJ, USA), and were soluble in water.

The lyophilized MEPs (VT1 and VT2) were dissolved in autoclaved Milli-Q water (Milli-QW) to prepare a concentration of 8 mg/mL and kept at −20 °C until use. Peptides were mixed 24 h before immunization with 8% Montanide™ gel 01 PR (Seppic, La Garenne Colombes, France) in a 40:60 ratio (*v*/*v*) and kept at 4 °C until use.

### 2.5. Standardization of Immunization Doses

For the dose optimization experiment, three groups (2 rabbits in each group) for each MEP were used. For both the MEPs, two dosages were used, and in each dose, 2 rabbits were immunized. Rabbits of the control group were injected with adjuvant mixed with the same volume of autoclaved Milli-QW (Table 1). A 500 µL of blood was collected from the saphenous vein at 0, 14, 28, and 42nd day of immunization using a vacutainer-containing clot activator for serum separation and the sera were stored at −20 °C until use.

### 2.6. Standardization of ELISA for Monitoring of IgG and IgM Response

All the variables were optimized by employing the checkerboard method [46]. Indirect ELISA was carried out to monitor serum IgG and IgM levels in the immunized in comparison to the control rabbits. The plates were coated with different concentrations of VT1 and VT2 antigen (0.5, 1, 2, 4 µg antigen/100 µL) diluted in carbonate–bicarbonate buffer (pH = 9.6) and incubated overnight at 4–8 °C in a humidified chamber. The plates were blocked using blocking buffer (1–5% *w*/*v* BSA) and kept in an incubator set at 27 °C for 2 h. Subsequently, 100 µL of sera samples (diluted at 1:100, 1:200, 1:400; 1;800 concentration in 0.1% *w*/*v* bovine serum albumin (BSA) in phosphate-buffered saline with 0.1% Tween 20 (PBS-T)) were added in each well and the plates were incubated at room temperature for 2 h with gentle continuous shaking. The HRP-conjugated anti-rabbit IgG and IgM (diluted at 1:2500, 1:5000; 1:10,000, 1:20,000 in blocking buffer) were added and incubated for 1 h at room temperature with gentle continuous shaking. A 100 µL of TMB (3,3′,5,5′-tetramethylbenzidine) substrate solution was added into each well and incubated at 27 °C for 30 min in the humidified chamber in the dark. The reaction was stopped by adding 50 µL of stop solution/well (12.5% H_2_SO_4_) and the absorbance was measured at 450 nm wavelength using an ELISA reader (BioRad, Hercules, CA, USA). Each concentration of antigen, sera, and adjuvant was tested in triplicate well for each antigen. All other controls were maintained to compare the data.

A plot of OD (*Y*-axis) vs. log sera concentrations (*X*-axis) for each antigen concentration (µg/mL) was drawn and an optimum concentration at which the highest signal-to-noise ratio that presentsan acceptable background was selected. The immunization dose was selected on the basis of maximum IgG and IgM response obtained in the dose optimization experiment.

### 2.7. In Vivo Immunization and Challenge Study

Based on the dose standardization experiment, doses for VT1 and VT2 were selected. For the challenge study, eighteen NZW rabbits were divided into three groups of six animals in each. Rabbits of the control group were injected with 8% Montanide™ gel 01 PR with Milli-QW in a 40:60 ratio (*v*/*v*). Rabbits were immunized (VT1 and VT2) with the selected doses given at14 day intervals.

All the rabbits of immunized and control groups were challenged on the 80th day of the first immunization with 14–18 day old *H. anatolicum* larvae (generated from 25 mg of eggs/rabbit). Subsequently, on the 112th day, the immunized and control rabbits were again challenged by 10 pairs of adults to evaluate the immunization efficacy [29,47].

The engorged nymphs dropped from each immunized and control rabbit were counted and kept at 28 °C and 80% humidity for moulting into adults [32]. The number of nymphs moulted to adults was compared between the immunized and control group of rabbits. Similarly, adult females dropped from each rabbit were counted and weighed. The individual adult was kept in a tick-rearing tube for oviposition. The feeding and reproductive parameters of adults that fed on immunized and control rabbits were compared to evaluate the efficacy of immunization. The immunization efficacy against larvae and adults were calculated using the standard formula (See Table 2) [30].

### 2.8. Immune Response Monitoring

After three inoculations of two concentrations of VT1 and VT2, ELISA was conducted as mentioned above, employing optimized conditions. The blood was collected from day 0 to day 112 post-immunization (DPI), sera were separated and processed for monitoring IgG and IgM responses at different time points. The mean optical density was calculated at each point in individual immunized and control groups of rabbits and analyzed statistically.

### 2.9. Expression Profile of Cytokines (IL-2, IL-4, IL-5, and INF-γ) by Q-PCR

Relative quantification of the targeted genes in immunized and control rabbits were measured by Q-PCR method. The primers for the targeted genes were designed using Primer3Plus online tool (presented in Appendix A). Total RNA was extracted from the PBMCs (peripheral blood mononuclear cell) of experimental animals using TRIzol reagent (Invitrogen, Waltham, MA, USA) as per the manufacturer’s protocol. The recovered RNA was quantified using Nanodrop. Freshly isolated total RNA was used for cDNA synthesis using iScriptTM cDNA synthesis kit (Bio-Rad, Hercules, CA, USA).

#### 2.9.1. Optimization of Q-PCR

The PCR conditions were standardized in 96-well one step Q-PCR machine (Applied biosytem, Waltham, MA, USA) for optimum annealing/extension temperatures, the presence of any nonspecific reaction such asself-primer dimer, cross-primer dimer, and hairpin. The reaction mixture was prepared for a 10 µL reaction using the following components: cDNA generated from 4 µg of total RNA, 2X SYBR green PCR master mix and 1pg both of the primers were incorporated. The reaction conditions were optimized with initial denaturation of 95 °C for 10 min followed by 40 cycles of denaturation at 95 °C for 15 s and annealing/extension at 52–54 °C for 1min.

#### 2.9.2. Relative Gene Expression Profile

For quantitative/real time measurement of the expression profile of Th1- type (IL-2, IFN-γ) and Th2 –type (IL-4, IL-5) cytokines, two step-qPCR was performed, keeping rabbit GAPDH as an endogenous control/reference. Quantitation-comparative (2^−∆∆Ct^) programs were used for relative quantitation with a reference (NTC) and endogenous control (GAPDH). MicroAmp fast optical plate (96-well) was used for running the reaction. Prepared cDNA, which was later taken as the template (4 µg of total RNA), was used in the reaction mixture along with SYBR^®^ Green master mix2X (5 µL), primers (1 pg each forward and reverse), and NFW (making final volume up to 10 µL). For each sample and control, reaction was kept in triplicate to minimize technical error. The reaction condition was initial denaturation at 95 °C for 10 min, followed by 40 cycles of denaturation at 95 °C for 15 s and annealing at 53 °C for 1 min.

#### 2.9.3. Quantification of Cytokines (IL-2, IL-4, and INF-γ) by ELISA

IFN-γ, IL-2, and IL-4 were quantified in serum samples collected on day 0, 80 (before challenge), and 82 (48 h after larvalchallenge) using ELISA kits (RayBiotech, Inc. Norcross, GA, USA) following the manufacturer’s protocol and the data were compared with the control group of rabbits.

### 2.10. Statistical Analysis

Relative quantification of gene expression was performedfollowing the method illustrated by Livak and Schmittgen [48]. For the normalization of gene expression, the geometric mean of the CT value of housekeeping gene, viz. GAPDH, was applied. One-way ANOVA and the Tukey test were used to compare the fold change of gene expression in the immunized group of rabbits in comparison to the control rabbits.

All the data were subjected to a statistical analysis using GraphPad Prism9. The *t*-test was carried out for comparing the OD values of sera collected from individual groups of rabbits and post-challenge entomological data collected from the immunized and control group of rabbits.

## 3. Results

### 3.1. Nature and Sub-Cellular Localization of the Selected Proteins

Upon an evaluation of the nature of selected proteins using SignalP 6.0 server, it is revealed that the targeted proteins except TPM have signal peptides(sec/SP1) with a single identified cleavage site. Using DeepLoc 1.0 server, the targeted proteins TPM, FER2, and VgR were predicted as soluble and are localized at the Golgi apparatus extracellularly and in plasma membrane, respectively (Appendix A).

### 3.2. Immunodominant Epitope Prediction and Their Characteristics

After meticulously screening a large number of B- cellepitopes using BepiPred-2.0 web and ABCpred servers, 29 B-cell epitopes were selected based on properties such asallergenic, antigenic, and physiochemical properties (Appendix A). Finally, the best two epitopes were selected (Table 3) based on their conservancy, surface accessibility (NetSurfP-2.0), and absence of transmembrane helices (TMHMM Server v. 2.0). The level of conservation of B-cell epitopes in different tick genera was 69 to 100%.

CTL epitopes were selected from three targeted proteins using NetMHCpan 4.1 (Allele of BoLA) and CTLpred servers. Seventy-one CTL epitopes were selected based on their binding affinity to different BoLA, allergenicity, and antigenicity (Appendix A). Out of seventy-one predicted epitopes, eight CTL epitopes were chosen for final constructs based on their conservancy, class-I immunogenicity score, and absence of transmembrane helices (TMHMM Server v. 2.0) in the epitopes (Table 3). A 43–100% conservation of protein among the different tick genera was observed in the selected epitopes.

Initially, a total of seventy helper T cell epitopes were selected using NetMHCIIpan 4.0 (HLA class II based) and IEDB MHC II servers (Appendix A). Finally, nine epitopes were selected based on their level of conservation, nature (exposed peptides), absence of transmembrane helices (TMHMM Server v. 2.0), and solvent accessibility (Table 3).

### 3.3. Prediction of Antigenic and Physiochemical Properties of the Constructs

The antigenicity of VT1 and VT2 constructs were 0.46 and 1.046, respectively, when compared with the parasite proteins (server can predict antigenicity of the unknown protein on the basis of viral, bacterial, or parasitic protein databases). Both the constructs were non-allergenic on both AllerTOP v2.0 and AllergenFP servers. Predicted pIvalues of VT1 and VT2 were estimated as 4.97 and 4.61, respectively, representing acidic in nature. The predictive instability index and other physiochemical parameters of VT1 and VT2 indicate stability of the constructs (Appendix A).

### 3.4. Predicted Secondary and Tertiary Structure

In the predicted secondary structure of the peptides, the percentage of alpha-helices (VT1-49.4%; VT2-75.86%) was higher than coils (VT1-43%; VT2-24.134%) followed by beta-strands (VT1-7.6%; VT2-0%) (Appendix A). However, the surface accessibility of VT1 was greater than VT2, which is attributable to the presence of a higher percentage (86%) of exposed residue in VT1 (including exposed and medium-exposed residue).

Similarly, the tertiary structures were generated using Raptor-X Property and SWISS-MODEL for VT1 and VT2, respectively (Figure 2A,B). The tertiary structure of VT1 was predicted based on PDB5b4xB as the best template with its *p*-value of 4.12 × 10^−5^. Similarly, the tertiary structure of the VT2 construct was generated by Raptor-X Property and SWISS-MODEL. The Swiss Model of VT2 was selected because of the better quality structure than Raptor-X. The models of VT1 and VT2 were refined and validated by ProSA. The validated VT1 and VT2 models were used for docking with different TLRs such asTLR-2, TLR-4, and TLR-9.

### 3.5. Refinement and Validation of the Tertiary Constructs

Out of all of the refined models, model 1 (first model on the server among the five refined models) was selected for both the MEPs. Essential parameters such as MolProbablity, RMSDclash score, poor rotamers, and Ramachandran plot were better in this case (Appendix A).

Moreover, Ramachandra plot analysis also signifies better structure analysis in VT2 than VT1 and it reveals a Ramachandran plot score of 98 and 88%, respectively (Appendix A). Further, ProSA-web shows a Z-score of −2.23 and −4.08 for VT1 and VT2, respectively, which lie within the score range that is commonly found in the case of native proteins of similar size (Figure 3A,B).

### 3.6. Molecular Docking and Molecular Dynamics Simulation

Molecular docking assesses the interactions between a ligand and a receptor molecule to determine the stability and binding affinity of their docked complex. The docking was significant in both the peptides with high negative energy values for top-ordered protein–protein docking complex (Appendix A, Figure 4A–F). The best scored docking models were selected based on amino acid residues involved in the interactions with the individual lowest energy score. Molecular interactions of the residues in the docked complexes were recognized by PDBsum (pictorial database of 3D structures in the Protein Data Bank) (Appendix A.)

The VT1 and VT2 constructs with TLR-9 produced a reliably stable simulation, as shown by normal mode analysis (NMA). The simulation study was conducted to specify the motions of molecules and atoms in the construct. The deformability graph of the complex depicts peaks in the graphs, which characterizes the regions of the protein with deformability (Figure 5A). The B-factor values derived by normal mode analysis are proportional to root mean square and they measure the uncertainty of each atom (Figure 5B). The B-factor value and root mean square deviation (RMSD) per residue amino acid showed more flexible residues with a sharp peak (Figure 6A,B) using MDWEB server. Better protein stability is indicated by a lower RMSD value, and the acceptable range for RMSD is between 0 and 1.2. The RMSD value for both VT1 and VT2 were in the range.

### 3.7. Dose Standardization on the Basis of IgG and IgM Responses

The optimum antigen concentration in ELISA was determined as 1.0 µg/mL and 500 ng/mL for VT1 and VT2, respectively. The 1:100 dilution of sera was optimized for both VT1 and VT2 while secondary antibody concentrations were determined as 1:5000 and 1:10,000 for IgG and IgM, respectively (Appendix A).

The IgG and IgM levels were higher in animals inoculated with 50 µg dose of VT1 than the 100 µg dose group (Table 4). A 4.6-fold (0.74 ± 0.01) and 7.7-fold increase (0.87 ± 0.01) in the IgM and IgG levels, respectively, was recorded at 42 DPI. However, in the case of VT2-immunized animals, a significantly higher IgG and IgM response was recorded in the animals of the100 µg total dose group in comparison to animals immunized with a dose of 50 µg (Table 5). Antibody responses in both the immunized groups of rabbits were significantly higher (*p* < 0.05) than in the control group. Based on the antibody levels, the in vivo trial dose was selected as 50 µg and 100 µg/rabbit for VT1 and VT2, respectively.

### 3.8. In Vivo Efficacy against Challenge Infestation

No observable clinical signs nor any inflammatory reactions were seen at the injection site in any of the rabbits after immunization with VT1 and VT2 antigens.

Against larvae

On average, larvae fed for 14 days in the controlanimals while a mean feeding duration of 16 days was recorded in the VT1-immunized rabbits. Yet, a substantial amount of 67.9% of fewer nymphs dropped from the VT1-immunized rabbits than from the control rabbits. (Table 5). Insignificant differences in the feeding period of larvae fed on VT2-immunized rabbits in comparison to the larvae fed on control rabbits werenoted. However, a significant rejection of 81.9% (DT%) in the number of nymphs dropped from VT2-immunized rabbits in comparison to ticks fed on the control rabbits was noted (Table 6; Appendix A). The moulting percent (MO%) of nymphs were 79.34 and 83.04 for VT1 and VT2, respectively. The efficacy (E%) of immunization was 93.3% for VT1 and 96.9% for VT2 and is presented in Table 6.

Against adults

The unfed adults started feeding within 24 h of release on the rabbit. After 12–14 days of feeding, engorged females started to drop from rabbits of all the groups. Significant differences in the number of dropped females were recorded amongst the groups. Further, significant differences in the engorgement weight and egg masses were recorded when the corresponding data were compared between the ticks fed on VT1, VT2, and control rabbits. The RI values of the engorged ticks showed significant differences (VT1—0.28; VT2—0.26; and control—0.51) among all the groups. The DT %, DR %, and E % were 43.6, 44.4, 89.9 and 43.6, 43.03, 86.4 for VT1- and VT2-immunized rabbits, respectively (Table 7).

### 3.9. Humoral Immune Responses to VT1 and VT2

On the day of immunization (day 0) all the rabbits showed a baseline IgG and IgM with an OD ranging from 0.09 to 0.12 and 0.16 to 0.19, respectively. After immunization, the rabbits immunized with VT1 showed significantly higher levels of anti-VT1 IgG (Figure 7) and anti-VT1 IgM (Figure 8) antibodies at 14, 28, 51, 84, and 112 DPI. The mean anti-VT1 IgG level rose to 2.8 and 8.6 times as compared to the control group on the 14th and 51st DPI. After the 51stDPI, a drop in IgG response was noted, but the value was still >5 times higher than the control group of animals at112 DPI.

The mean IgG OD value of the sera from VT2-immunized rabbits rose to 2.8 and 8.3 times as compared to the control rabbits on the 14th and 28th DPI. A maximum VT2-specific IgG response was observed at 28 DPI. The response was reduced on the 51st day and maintained until112 DPI (Figure 7). The IgG response was higher in all the immunized rabbits in comparison to the control group of animals throughout the experiment.

The mean IgM OD values of the sera collected on the 14th and 51st DPI were 0.27 ± 0.01, 0.49 ± 0.01 and 0.59 ± 0.021, 0.21 ± 0.01 in VT1 and VT2 groups, respectively, and the values were nearly 1.5–3.3 and 1.4–2.7 times higher in VT1- and VT2-immunized rabbits, respectively, in comparison to control. A fall in IgM response after 51 and 14 DPI was noted in VT1- and VT2-immunized animals, respectively (Figure 8).

### 3.10. Cellular Immune Responses to VT1 and VT2

MEP-induced T-cell responses on day 0, 80 (before challenge), and 82 (48 h after larval challenge) were determined by quantification of cytokines (IFN-γ, IL-2 and IL-4) immunized compared with the response observed in control rabbits. No significant difference was noticed in the mean IFN-γ level between the immunized and control rabbits. However, IL-2 level was significantly reduced in VT2-immunized rabbits after challenge infestations as compared with the pre-challenge level (unpaired *t*-test; *p* < 0.05). Similarly, a reduction in IL-4 level (35.66 µg/mL) was also observed in VT2-immunized animals as compared with the responses observed in the control rabbits (47.81 µg/mL). In contrast, the mean IL-2 level in the VT1 group of rabbits was 40.59 pg/mL as compared with the pre-challenge level of 28.31 pg/mL. The mean IL-4 level was significantly (*p* < 0.01) reduced in VT1-immunized (32.90 µg/mL) rabbits as compared with the control rabbits (47.81 µg/mL) (Figure 9A–F).

Relative gene expression was also determined by Q-PCR in terms of fold change. Comparatively, a higher level of (*p* < 0.001) IL-2 gene expression was observed in both VT1- and VT2-peptide-immunized rabbits. There was also a significant 20-fold increase in the IFN-γ level in VT2-immunized rabbits than the control rabbits. A significant reduction of the IL-5 level was recorded in both the immunized groups of rabbits. The comparative cytokines transcript level is shown in Figure 10A–H.

## 4. Discussion

Willadsen [49] proposed that multiple antigens targeting various functions of ticks have a better potential to be successful in field conditions. Recently, one multi-antigen-based m-RNA vaccine formulation against *Ixodes scapularis* has been tested on guinea pigs [50]. Immunization with 19ISP-based mRNA vaccine followed by a challenge with *I. scapularis* revealed the formation of erythema in animals following a tick bite. The feeding period of challenged ticks was reduced, resulting in a reduction in the engorgement weights. Similarly, Jelnková et al. [51], by employing an immunoinformatics method, identified the junctional region of the circumsporozoite protein, CIS43 VLP of *Plasmodium falciparum*. Immunization of mice and challenge with one thousand sporozoites have resulted in ~72–90% reduction of liver-stage parasite burden in mice.

Similarly, in the present study, an effort was made to design MEP constructs using an immunoinformatics approach which utilizes different algorithms to predict potential epitopes. The immunodominant peptides were picked based on functional properties reported by Manjunathachar et al. [30] and Xavier et al. [31]. It has been reported that tick salivadown-regulates the Th1 cytokines IL-2 and IFN-γ and up-regulates the Th2 cytokines IL-4, IL-5, IL-6, and IL-10 during feeding [52,53,54]. Thus, to combat such immune-modulation events within hosts, we designed two sets of MEP constructs incorporating functional epitopes. To provide proper proteasomal cleavage sites for different immune cells, MHC-I, MHC-II, and linear B-cell epitopes were linked together using GPGPG linkers [55].

An analysis of the secondary structure of the final construct demonstrated that the proteins consisted pre-dominantly of alpha-helices (VT1-49.4 and VT2-72.86%) followed by coils and then beta-strands, respectively. In nature, the unfolded protein portion and alpha-helical coils have been reported to be important forms of “structural antigens” [56]. The VT2 construct showed more than 72% alpha-helices, indicating a promising structural antigen. The 3D structure of the final constructs strikingly improved after refinement with GalaxyRefine server and showed desirable results on Ramachandran plot predictions. The Ramachandran plot shows that most of the residues are in the favored and allowed regions (VT1-88%, VT2-98% and VT1-8% and VT2-2%) with very few residues (VT1-4% and VT2-0%) in the outlier region, the sign of a satisfactory model. Although the overall final tertiary structure quality of the VT2 peptide was better than VT1 in terms of most of the predicted physicochemical properties, the overall Ramachandran plot predictions for both the final models were satisfactory as it is highly desirable to obtainmaximal residues which liein the favored regions followed by allowed regions, while minimal residues are in the disallowed regions [57].

Docking of vaccine constructs were performed with TLRs that show favorable docking. The docking results were further validated by simulation studies. MD simulation study was carried out to attain information on construct stability using the MDWEB server. The residue-based RMSD plot peaked at more flexible residues, such as L14, G33, and Y38 in VT1. Both structures fluctuated, indicating their considerable flexibility and validating it as suitable vaccine structures. Similarly, Atapour et al. [58] recorded a high degree of fluctuation in the residues at positions 219 and 617 in a *P. falciparum* multi-epitope vaccine. The results obtained in the form of B-factor and deformability plots indicated that the molecular structures of both of the peptides were stable, and the overall stability of the VT2 constructs was higher compared to the VT1.

At the initial phase of any infection, the TLRs trigger innate immune responses after the recognition of pathogen-derived compounds/molecules. The TLRs are type-I integral membrane receptors, with an N-terminal domain for the recognition of ligands and the cytoplasmic signaling domain at the C-termini [59]. All the TLRs are made up of a special type of hydrophobic residue at certain intervals, which is typically 22–29 residues of leucine-rich repeat (LRR). This hydrophobic LRR portion is the binding site for ligands. Several specific compounds have been identified in different tick saliva, which interfere with the functioning and activity of dendritic cells. These tick salivary proteins promote IL-10 by bone marrow-derived DCs (dendritic cells) in response to TLR- 2, TLR-4, and TLR-9 ligands [60,61]. Accordingly, in the present study, VT1 and VT2 constructs were docked with TLR-2, TLR-4, and TLR-9 using Haddock 2.4 server. In addition, one recent study has indicated that the S protein of SARS-CoV binds to TLR4 directly and activates monocytes and neutrophils [62]. In the present docking study, a similar type of molecular interaction between peptides with the LRR portion of TLRs was observed, which indicated significant innate immune responses. Among the three TLRs, the TLR-9 had the best-docked model with both the peptides having the lowest RMSD score (0.5). The lowest energy score of these complexes reflected the highest binding affinity between TLR-9, VT1, and VT2 peptides. The NMA (normal mode analyses) results confirmed that the final construct can interact with TLR-9 suitably [63].

Classical whole antigen-based vaccines have several detrimental effects such as an autoimmune response or allergic reactions. Synthetic peptide-based vaccines can be an alternate choice to overcome it. However, these peptides have low immunogenicity, and thus additional immuno-stimulating agents such as adjuvant or carrier molecules [64] are to be included. In the present study, the VT1 construct consisted of Fer2 (HTL), TPM (CTL/INF-γ-inducing), and VitR (HTL) epitopes linked with the GPGPG linker. Finally, to potentiate an immune response, a universal T helper cell epitope, the Pan DR epitope peptide (PADRE, AKFVAAWTLKAAA) was fused [65]. Previously, Almazán and coworkers [66] reported the fusion of the Pan DR epitope peptide with two *I. ricinus*-specific neuropeptides. Both the neuropeptides (SIFamide; SIFa) and myoinhibitory peptide (MIP) produced a significant level of IgG response at 73 DPI.

The VT2 was constructed with one B-cell epitope (INF-γ-inducing) of Fer2 and two immunodominant TPM epitopes (CTL and HTL/B-cell). Finally, to potentiate CMI and humoral immunity, CPP was fused at the C terminal end of the VT2. This CPP has been recommended as an efficient mediator for vaccine delivery as it may improve the uptake of antigens by APCs and is considered as a classical adjuvant [67].

Although several candidate proteins have been identified as potent vaccine candidates in different tick species [68,69,70], FER2 and TPM are mostly studied as they are highly conserved among different tick species and are present in all developmental stages [68,69]. Previously, Kumar et al. [29] and Manjunathachar et al. [30] reported 51.7% and 63.7% protection against larvae and 51.2% and 66.4% protection against adults of *H. anatolicum* challenge following immunization of cross-bred calves by rHaFER2 and rHaTPM, respectively.

The antigen dose and adjuvants are the two most important parameters to decide for a better immune response. In the present study, two dosages were tested with 8% Montanide™ gel 01 PR to stimulate both humoral (HI) and cellular immune (CMI) responses. This adjuvant has the potential to stimulate both the immune systems following immunization with peptide-based vaccine [71]. It produced a specific anti-P0 (P0 peptide) antibody in dogs and conferred 85% efficacy against *R. Sanguineus* challenge infestations. It is also easy to reconstitute the immunization dose using the adjuvant and is cost-effective as a veterinary vaccine with low adverse reactions, as described by Guzman et al. [71]. Similarly, in the present study, a significantly high level of anti-VT1 and VT2 antibody (IgM and IgG) responses were noted at 50 µg/animal of VT1 and 100 µg/animal dose of VT2 and was maintained until the feeding of the challenged stages of the tick species.

In the present work, immunization with VT1 confers strong immunity in rabbits, resulted in quick larval detachment, delayed tick feeding, low engorgement weights (0.55 times), and reduced egg masses (0.31 times) with an overall efficacy (E%) of 93.3% and 89.9% against larvae and adults, respectively. Rabbits inoculated with VT1-induced robust antibody (IgG) response and multiple T cell-related inflammatory cytokines and thus indicated the role of humoral and cell-mediated immunity in conferring protection against challenge infestations.

An increased anti-VT2 IgG level (IgG) and pro-inflammatory cytokine (IL-2) and decreased anti-inflammatory cytokine IL-4 are most likely responsible for conferring a significant (*p* < 0.05) level of protection against *H. Anatolicum* larval (96.9%) challenge in terms of the reduction in percentages of healthy and fully replenished nymphs dropped, along with an increased feeding period of ticks fed on the immunized group of animals in comparison to ticks fed on a control group of rabbits. An overall efficacy (E%) of 86.4% was obtained against adult challenge in terms of rejection in the number of attached and dropped ticks (DT% = 43.7), tick weight (DR% = 43.0), and egg masses (DO% = −54.1).

The importance of inducing both HI and CMI responses in conferring immunity against challenge infestations/infection is well documented [50]. Likewise, in this study, MEPs were designed to target both the CMI and HI responses. In the post-challenge period, a significant (*p* < 0.001) reduction in the anti-inflammatory cytokine (IL-4) expression was observed in the VT1- and VT2-immunized group of rabbits as compared with the response observed in the control group of rabbits. This finding was contradictory to the results obtained by Sajid and co-workers [50], where anincreased IL-4 level was reported in the immunized group as compared with the control group of animals. The down regulation of the Th1 cytokines IL-2 and IFN- γ and the up-regulation of the Th2 cytokines IL-4, IL-5, IL-6, and IL-10 during infestation is a common feature for many tick-host relationships [56,57,58]. A significant up-regulation of Th1 cytokines INF-γandIL-2 and down-regulation of Th2 cytokine IL-5 wasrecorded in the VT2-immunized group after 48 h of larval challenge. However, up-regulation of IL-2 and down-regulation of theTh2 cytokine IL-5 wasrecorded in the VT1-immunized group after 48 h of larval challenge. Similar results were demonstrated by Sajidet al. [50], where pro-inflammatory cytokine expression was enhanced in the 19ISP-based mRNA vaccine. It enhanced the recognition of tick bites and reduced *I. Scapularis* engorgement and prevented transmission of Lyme disease. Another peptide-based vaccine against mosquito has already been under phase 1 trial on49 healthy human volunteers. The salivary protein *A. gambiae* (AGS-*v)* has been reported to producea significant IgG antibody response in addition to a significant increase in INF-γ levels expression by the PBMCs at 42 DPI [72].

Although the minimum number of rabbits in place of cattle for immunization was used as a pilot experiment, the results obtained clearly demonstrated the suitability of MEP-based vaccine for a higher efficacy against challenge infestations.

## Figures and Tables

**Figure 1 vaccines-11-00881-f001:**
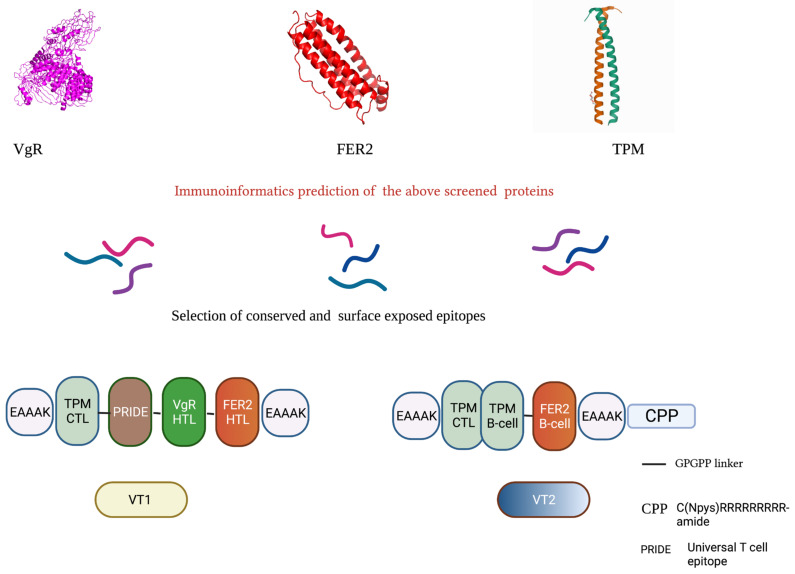
Schematic diagram of multi-epitopic peptide construct (created with biorender.com (accessed on 1 April 2022)). The VT1 contains TPM CTL epitope (KIVELEEEL), VitR HTL epitope (YGEPFLLYMLPNQIR), and FER2 HTL epitope (DTGLGEFLLDQQLRT) as indicated. The VT2 contains TPM CTL (KIVELEEEL), TPM B-cell epitope (LEEELRVVGNNLKSL), and one FER2 B-cell epitope (DFLEQEFLAEQVKSID) linked with GPGPG linker.

**Figure 2 vaccines-11-00881-f002:**
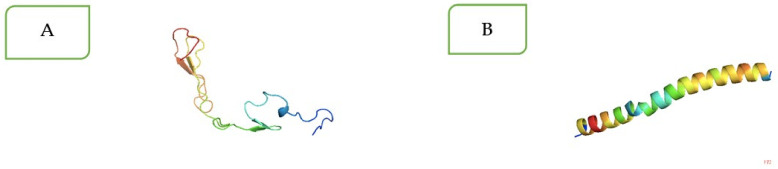
(**A**,**B**) The tertiary structure of the designed MEP construct VT1 (**A**) and VT2 (**B**). The VT1 3D structure modeling by Raptor-X based on the template with PDB ID 5b4xB, and the VT2 3D structure model is based on SWISS-MODEL.

**Figure 3 vaccines-11-00881-f003:**
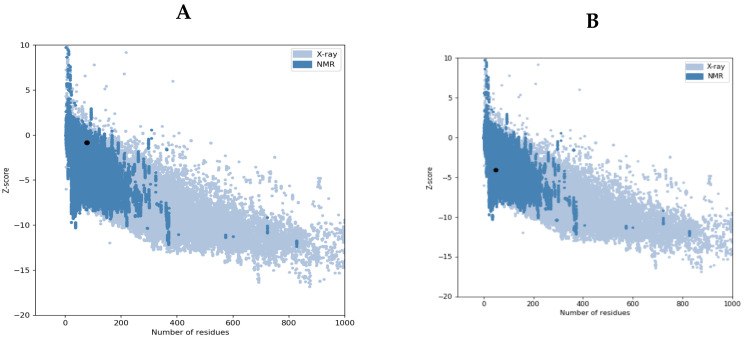
(**A**,**B**) VT1 and VT2 3D structures validation by ProSA-web. The Z-score of −2.23 and −4.08 for VT1 (**A**) and VT2 (**B**), respectively, which lie inside the score range.

**Figure 4 vaccines-11-00881-f004:**
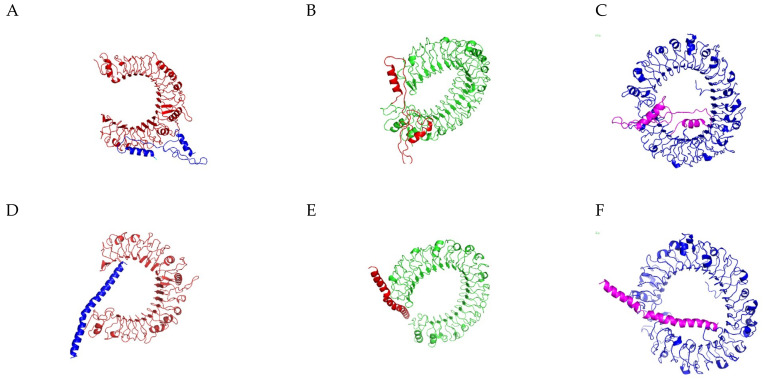
(**A**–**F**) Docked complexes of TLR-2, TLR-4, and TLR-9 with VT1 (**A**–**C**) and VT2 (**D**–**F**) peptides.

**Figure 5 vaccines-11-00881-f005:**
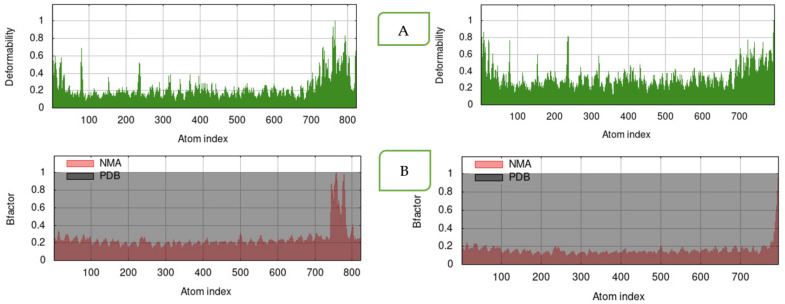
The results of molecular dynamics simulation of VT1 (left) and VT2 (right) constructs, and TLR-9 docked complex. (**A**) deformability; (**B**) variance (red color indicates individual variances and green color indicates cumulative variances) (iMods).

**Figure 6 vaccines-11-00881-f006:**
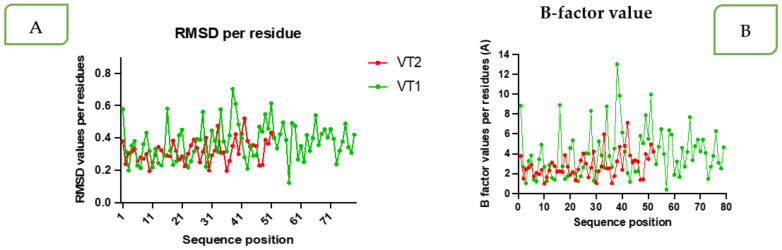
RMSD per residue (**A**) and B-factor value (**B**) per amino acid showing more flexible residues with the sharp peak (red for VT2 and green for VT1).

**Figure 7 vaccines-11-00881-f007:**
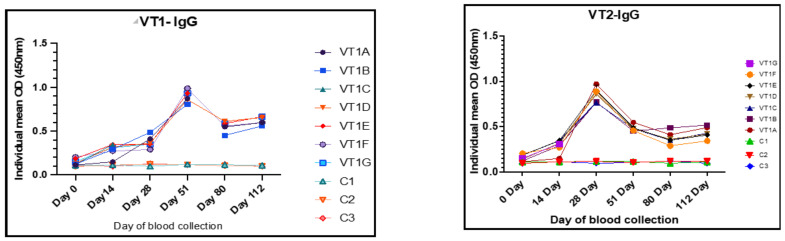
Humoral immune responses (IgG) on different time points in VT1- and VT2-immunized rabbits at doses of 50 µg and 100 µg, respectively. Different color dots represent mean of the individual animal (A-G) for immunized and C (1–3) for each control animal.

**Figure 8 vaccines-11-00881-f008:**
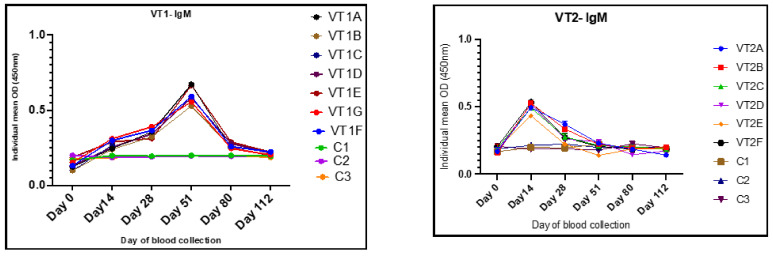
IgM responses on different time points in VT1- and VT2-immunized rabbits inoculated with 50 µg and 100 µg of antigen/animal, respectively. Different color dots represent mean of the individual animal (A-G) for immunized and C (1–3) for each control animal.

**Figure 9 vaccines-11-00881-f009:**
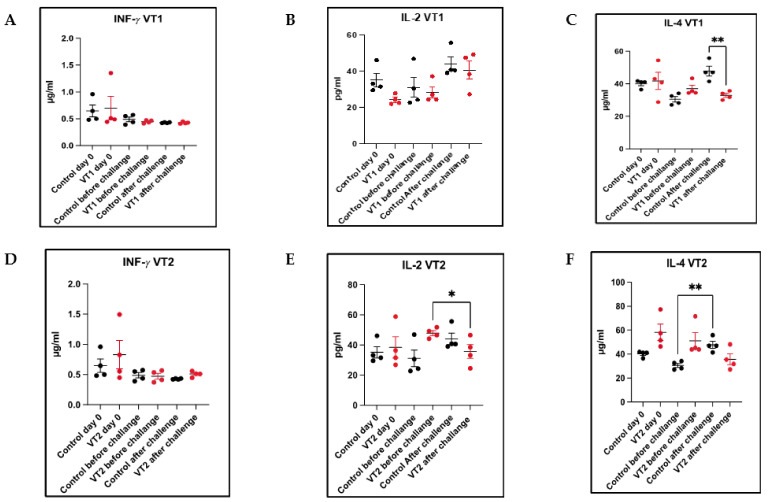
(**A**–**F**) Cellular immune responses in VT1- and VT2-immunized rabbits following inoculation of 50 µg and 100 µg of antigen/animal, respectively. The INF-γ (**A**,**D**), IL-2 (**B**,**E**), and IL-4 (**C**,**F**) responses to VT1 and VT2 antigens at different time intervals in immunized and control rabbits. Significant at * *p* < 0.05; ** *p* < 0.01.

**Figure 10 vaccines-11-00881-f010:**
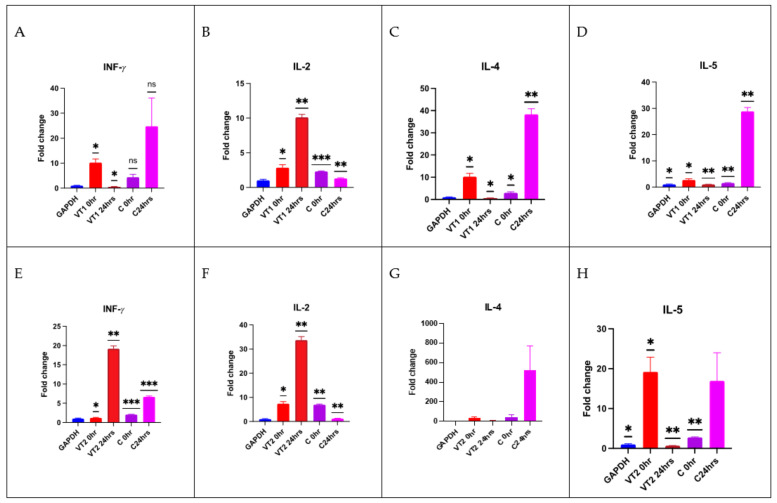
(**A**–**H**) Relative fold changes of INF-γ (**A**,**E**), IL-2 (**B**,**F**), IL-4 (**C**,**G**), and IL-5 (**D**,**H**) genes in different time intervals in VT1 (50 µg antigen/animal) and VT2 (100 µg antigen/animal) and control group of rabbits. Significant at * *p* < 0.05; ** *p* < 0.01; *** *p* < 0.001.

**Table 1 vaccines-11-00881-t001:** Immunization schedule, dose, adjuvant used, route of immunization of rabbits using VT2 and VT1 peptides.

Schedule of Immunization	VT2 * #		VT1 * #		Control
	Group 1	Group 2	Group 1	Group 2	
0 day	50 µg in 100 µL Mili-QW + 150 µL adjuvant	25 µg in 100 µL Mili-QW + 150 µL adjuvant	50 µg in 100 µL Mili-QW + 150 µL adjuvant	25 µg in 100 µL Mili-QW + 150 µL adjuvant	100 µL Mili-QW + 150 µL adjuvant
14 day	25 µg in 100 µL Mili-QW + 150 µL adjuvant	12.5 µg in 100 µL Mili-QW + 150 µL adjuvant	25 µg in 100 µL Mili-QW + 150 µL adjuvant	12.5 µg in 100 µL Mili-QW + 150 µL adjuvant	Same dose
28 day	Same as booster dose	Same as booster dose	Same as booster dose	Same as booster dose	Same dose

* 8% Montanide™ gel 01 PR as adjuvant: # i.m.

**Table 2 vaccines-11-00881-t002:** Formula of determination of vaccine efficacy against *Hyalomma anatolicum*.

For larvae	DT (%) = 100 (1 − NTV/NTC)	Where DT% is the percentage rejection in the number of nymphs; NTV, the number of nymphs dropped from the immunized rabbits; and NTC, the number of nymphs dropped from thecontrol rabbits.
	MO (%) = 100 (1 − MLI/MLC)	Where MO (%) is the percent reduction in moulting of engorged nymphs; MLI and MLC, the number of engorged nymphs moulted to adults from immunized and control rabbits, respectively.
	E (%) = 100 [1 − (CRT × CRM)]	Where E (%) is the percentage efficacy of antigen against larvae; CRT is reduction in the number of nymphs (NTV/NTC); CRM is reduction in number of engorged nymphs moulted to adults (MLI/MLC).
For adults	DT% = 100(1 − NTV/NTC)	Where DT% is the percentage rejection in the number of females; NTV, the number of females dropped from the immunized animals; NTC, the number of females dropped from the control animals.
	DR (%) = 100(1 − PMTV/PMTC)	Where DR (%) is the percentage reduction of mean weight of adult females; PMTV is the mean weight of adult females dropped from the immunized animals; PMTC is the mean weight of adult females dropped from the control animals.
	DO (%) = 100(1 − PATV/PATC)	Where DO (%) is the percentage reduction of mean weight of eggs; PATV is the mean weight of eggs of females which fed on vaccinated animals; PATC is the mean weight of eggs of females which fed on control animals.
	RI = Egg weight/engorge tick weight	Where RI is designated as reproductive index.
	RF (%) = 100(1 − RIV/RIC)	Where RF (%) is the percentage reduction in adult fertility; RIV is the mean RI of adult females dropped from the immunized animals; RIC is the mean RI of adult females dropped from the control animals.
	E (%) = 100 [1 − (CRT × CRO × CRI)]	Where E% is the percentage efficacy of antigens; CRO is reduction in egg laying capacity (PATV/PATC); CRT is the reduction in the number of adult females (NTV/NTC); CRI is the reduction in tick fertility.

**Table 3 vaccines-11-00881-t003:** Final shortlisted selected epitopes.

Protein	CTL Epitopes	IFN-γ Epitope	HTL Epitopes	IFN-γEpitope	B-Cell Epitopes	IFN-γEpitope
VgR	GVHVYHPVL (522)	Yes	**YGEPFLLYMLPNQIR** (1301)	No		
	ALFEDWLYW (1514)	Yes				
	ALLVLGYVL (1682)	No	LCVALLVLGYVLYRR (1680)	No		
FER2	INLELHASL (42)	No	NLELHASLVYMQMAA (43)	No	**DFLEQEFLAEQVKSID** (111)	Yes
			DDDPQMADFLEQEFL (141)	No		
			**DTGLGEFLLDQQLRT** (176)	No		
			LNAIPVSPQTNLFYS (33)	No		
TPM	RMDGLEGQL (140)	No	ELRVVGNNLKSLEVS (196)	No	**LEEELRVVGNNLKSL** (193)	No
	**KIVELEEEL** (189)	Yes				

Note: Selected epitopes are in bold.

**Table 4 vaccines-11-00881-t004:** Antibody responses before and after the second booster dose of VT1 antigen in immunized and control rabbits.

	IgG			IgM		
Dose	50 µg	100 µg	Control	50 µg	100 µg	Control
Day 0	0.11 ± 0.00 **	0.12 ± 0.01	0.12 ± 0.00	0.11 ± 0.00	0.13 ± 0.00	0.16 ± 0.03
Day 14	0.151 ± 0.01 **	0.23 ± 0.014	0.13 ± 0.00	0.32 ± 0.00	0.29 ± 0.00	0.18 ± 0.02
Day 28	0.412 ± 0.00 **	0.31 ± 0.01	0.13 ± 0.00	0.40 ± 0.00	0.40 ± 0.00	0.17 ± 0.00
Day 42	0.87 ± 0.01 **	0.57 ± 0.01	0.11 ± 0.00	0.74± 0.01	0.57 ± 0.00	0.17 ± 0.00

Significant at ** *p* < 0.01.

**Table 5 vaccines-11-00881-t005:** Antibody responses before and after the second booster dose of VT2 antigen in immunized and control rabbits.

	IgG			IgM		
Dose	50 µg	100 µg	Control	50 µg	100 µg	Control
Day 0	0.12 ± 0.01 *	0.11 ± 0.00	0.11 ± 0.00	0.17 ±0.03 *	0.12 ± 0.01 *	0.17 ± 0.01
Day 14	0.15 ± 0.01 *	0.29 ± 0.00	0.11 ± 0.00	0.38 ± 0.03 *	0.53 ± 0.02 *	0.2 ± 0.00
Day 28	0.47 ± 0.01 *	0.97 ± 0.01	0.12 ± 0.01	0.27 ± 0.02 *	0.34 ± 0.01 *	0.19 ± 0.00
Day 42	0.45 ± 0.01 *	0.85 ± 0.01	0.12 ± 0.02	0.21 ±0.01 *	0.25 ± 0.03 *	0.2 ± 0.01

* Significant at *p* < 0.05.

**Table 6 vaccines-11-00881-t006:** Protective efficacy of VT1 and VT2 peptides against challenge infestations of larvae of *H. anatolicum* (IVRI-2 strain).

Group	Mean wt (mg)/Nymph	No. of Nymphs (Mean ± SE)	DT%	MO%	E%
VT1	7.67 ± 0.0	72.75 ± 13.5 *	67.9	79.3	93.3
VT2	7.11 ± 0.0	41 ± 2.6 ***	81.9	83.0	96.9
Control	8.13 ± 0.0	226.7 ± 12.4 ***			

* Significant at *p* < 0.05; *** *p* < 0.001. DT% = percent rejection of challenged larvae in comparison to control; MO% = percent reduction of molting percentage of larvae to nymph; E% = efficacy percentage.

**Table 7 vaccines-11-00881-t007:** Entomological parameters of ticks that fed on VT1- and VT2-immunized group of rabbits.

Group	No. of Females Dropped (Mean ± SE)	wt (mg)/Tick (Mean ± SE)	Egg Masses (Mean ± SE)	DT%	DR%	DO%	RF%	E%
VT1	8 ± 0.4 ***	196.7± 6.8 ***	57.0 ± 6.7 ***	43.7	44.4	68.5	42.7	89.9
VT2	8 ± 0.4 ***	202.2± 4.7 ***	82.2 ± 6.5 ***	43.7	43.0	54.2	47.5	86.4
Control	14.2 ± 0.4	357.3 ± 4.3	181.4± 3.4					

Significant at *** *p* < 0.001. DT% = percent rejection of challenge adults in comparison to control. DO% = percent reduction in egg masses; DR% = percent reduction of mean weight of adult females; RF% = percent reduction in adult fertility; E% = efficacy percentage.

## Data Availability

Data will be made available on request.

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
