# Peer review of "Protective Efficacy of Multiple Epitope-Based Vaccine against Hyalomma anatolicum, Vector of Theileria annulata and Crimean–Congo Hemorrhagic Fever Virus"

_vaccines, 2023, doi:10.3390/vaccines11040881_

Round 1

Reviewer 1 Report

The work by Nandi et al. designed multi-epitope vaccines against Hyalomma anatolicum which is an important vector for many animal and zoonotic pathogens. They analyzed epitope info of several target proteins by bioinformatic approaches, constructed two multi-epitope peptides and tested their efficacy, antibody levels and cytokine responses after immunization, with both MEPs showing good protective efficacy. The work in general is well organized, and data are shown clearly. therefore, I only have a few points concerned:

Major points:

 Notably, as ectoparasites, how hyalomma vaccines work might be quite different with typical vaccines. The specific antibody after immunization may block normal functions of the parasite proteins, especially in the case of target proteins like FER2, thus impairing the development of the organism or egg, in different with direct protective effects by the host immune responses in typical vaccinations. Simply measurement of antibodies or cytokine levels may not be the best way to evaluate how good is the vaccine as the vaccine might work by a different way. 

Minor points:

1. In figure 1, according to the (predicted) protein structure of Vitr/Fer2/Tropo, do the antigen peptides used in these studies locate on the surface of these proteins? This info might indicate whether the peptides are linear epitopes, and may also give some hints on the maintenance of the original structure of these peptides when they are taken out for new constructs. 

2. In Table 2, what do the letter ‘P’ and ‘N’ stand for? This should be explained.

3. All supplementary tables are missing.

4. Line 486, while in the text the authors explained the VT1 dose used are 50 μg and 100 μg and the 50 μg group performs better. But in both Table 4, 5 and the Supplementary Tables 25 μg and 50 μg groups are shown. Please check the numbers and text description.

Author Response

Major points:

 Notably, as ectoparasites, how hyalomma vaccines work might be quite different with typical vaccines. The specific antibody after immunization may block normal functions of the parasite proteins, especially in the case of target proteins like FER2, thus impairing the development of the organism or egg, in different with direct protective effects by the host immune responses in typical vaccinations. Simply measurement of antibodies or cytokine levels may not be the best way to evaluate how good is the vaccine as the vaccine might work by a different way. 

Response : 

We are thankful to the reviewer to raise the points. We know there are other possible mechanisms which are working in synergistic fashion to provide protection against challenged ticks. However, since the inception of the subject till date, monitoring of antibody response post immunization of hosts using tick antigen is considered ideal to correlate the protection. We have conducted a good number of study and found there is a positive correlation between antibody response and protection conferred. The experiment is organized in the same manner and the result is presented, please.

Comments :

Minor points:

In figure 1, according to the (predicted) protein structure of Vitr/Fer2/Tropo, do the antigen peptides used in these studies locate on the surface of these proteins? This info might indicate whether the peptides are linear epitopes and may also give some hints on the maintenance of the original structure of these peptides when they are taken out for new constructs. 

Response:

Yes, all the predicted epitopes used in this study were exposed to generate strong immunity.

In the present study, all predicted epitopes were linear. To maintain functionality and stability of the construct GPGPG linker and EAAAK were used.

Comments: In Table 2, what do the letter ‘P’ and ‘N’ stand for? This should be explained.

Response: Described after table

Comments: All supplementary tables are missing

Response :Included please

Comments: 

Line 486, while in the text the authors explained the VT1 dose used are 50 μg and 100 μg and the 50 μg group performs better. But in both Table 4, 5 and the Supplementary Tables 25 μg and 50 μg groups are shown. Please check the numbers and text description.

Response: A major mistake and the same has been corrected throughout the ms. 

Reviewer 2 Report

The article "Protective efficacy of multiple epitope-based vaccine against Hyalomma anatolicum, vector of Theileria annulata and Crimean-Congo haemorrhagic fever virus" presents a valuable scientific contribution to the field of vaccine development for the control of tick infestations. The findings of the study are sound and may be helpful to improve the livestock productivity and health. All is OK about the article except it is written in such a way that the message or language is not much clear and straightforward in addition to many grammatical and typo errors which need extensive rephrasing of the text.    

Author Response

Comments:The article "Protective efficacy of multiple epitope-based vaccine against Hyalomma anatolicum, vector of Theileria annulata and Crimean-Congo haemorrhagic fever virus" presents a valuable scientific contribution to the field of vaccine development for the control of tick infestations. The findings of the study are sound and may be helpful to improve the livestock productivity and health. All is OK about the article except it is written in such a way that the message or language is not much clear and straightforward in addition to many grammatical and typo errors which need extensive rephrasing of the text.

Response: We are thankful to the reviewer for the suggestions. The entire ms has been thoroughly checked, mistakes were corrected,. necessary figures and tables are properly presented, and references have been matched with the text. 

Reviewer 3 Report

The manuscript “Protective efficacy of multiple epitope-based vaccine against Hyalomma anatolicum, vector of Theileria annulata and Crimean-Congo haemorrhagic fever virus” by Abhijit Nandi et al is an investigation that includes the design of a peptide-based vaccine containing multiple epitopes for the prevention of Hyalomma anatolicum, an invertebrate vector of diverse diseases in cattle. The experimental approach included the identification of the sequence of immunodominant peptides, the peptide design, the subsequent bioinformatic analysis in structural and functional terms for the chosen purposes and, finally, the synthesis and evaluation in a rabbit model. The results obtained are promising as an initial stage for its future evaluation in the organism of interest.

The presentation of the manuscript is correct, and it is well structured. Perhaps, some subheadings of the methodology could be condensed (e.g., those related to bioinformatics analyzes of peptides).

Other comments:

*Line 23. It would be convenient not to open the parenthesis twice without first closing it. The use of square brackets is recommended.

*Did the authors carry out some type of biochemical analysis on the peptide samples? I understand that the company in which the synthesis was carried out provided more details of quality and methodologies for its determination that could be added to the corresponding section (lines 185-191).

*It could be useful to add a paragraph in the discussion section where the existence of similar vaccines in the veterinary market be analyzed, and if the costs of their industrial production would make the proposal feasible.

Author Response

The manuscript “Protective efficacy of multiple epitope-based vaccine against Hyalomma anatolicum, vector of Theileria annulata and Crimean-Congo haemorrhagic fever virus” by Abhijit Nandi et al is an investigation that includes the design of a peptide-based vaccine containing multiple epitopes for the prevention of Hyalomma anatolicum, an invertebrate vector of diverse diseases in cattle. The experimental approach included the identification of the sequence of immunodominant peptides, the peptide design, the subsequent bioinformatic analysis in structural and functional terms for the chosen purposes and, finally, the synthesis and evaluation in a rabbit model. The results obtained are promising as an initial stage for its future evaluation in the organism of interest.

The presentation of the manuscript is correct, and it is well structured. Perhaps, some subheadings of the methodology could be condensed (e.g., those related to bioinformatics analyzes of peptides).

Comments: Line 23. It would be convenient not to open the parenthesis twice without first closing it. The use of square brackets is recommended.

Response: Changes are done as per the recommendation

Comments: Did the authors carry out some type of biochemical analysis on the peptide samples? I understand that the company in which the synthesis was carried out provided more details of quality and methodologies for its determination that could be added to the corresponding section (lines 185-191).

Response: 

The HPLC purity and Mass Spec results were provided by the company.

Corrected molecular weight and purity of the peptides are incorporated, please.

Comments: It could be useful to add a paragraph in the discussion section where the existence of similar vaccines in the veterinary market be analyzed, and if the costs of their industrial production would make the proposal feasible.

Response: No peptide vaccine of veterinary use is available in the market 

Reviewer 4 Report

The major defect in this paper is its presentation. It is extremely difficult to follow and the paper -- especially the results -- sometimes reads like random numbers and acronyms. The authors should do a major overhaul of the paper and probably delete a lot of the modeling studies. Figure legends and table legends are generally horrendous. The authors need to slowly and carefully go through the manuscript and think about it from a potential reader's perspective and clearly describe the experiments and their results, as well as the significance of those results. 

The abstract is too long, too technical, and too methodological. It could easily be cut to half its current size.

First sentence of introduction should probably be expanded to at least two sentences. 

Line 75: Three synthetic peptides () derived from BM86 of R. microplus ...

Line 84: delete comprising

Line 91: In addition is probably better the Besides

Line 112: delete Two different servers, 

Line 116: I would move the names of the two servers from line 117 to replace 'a specific server' in line 116

line 132: first use of MEP, but defined on line 180. There are numerous cases of non-standard abbreviations throughout the manuscript that are not defined

line 167: delete , if any

line 203: either delete the sentence or move it to the next section

line 332: change are having to have

line 334: plasma membrane is probably better than cell membrane

Table 2 needs a legend (what is P and N?)

line 354: 43-100%

line 373: what are the parasite proteins?

line 374: predicted pI values of VT1 and VT2 were

Figure 1: legend needs more information including an indication of which specific epitopes from Table 2 were used

Figure 3 has some serious formating issues and the legend is incomprehensible. Furthermore, is the figure really necessary? 

line 418: What is model 1?

Section 3.8. What is the point of this section? What is the signifcance? Is Figure 4 necessary? What is Figure 4 showing and how does it related to VT1 and VT2 as potential vaccines? 

line 447: What is RMSD and how does it relate to the immungenecity of VT1 and VT2. Is section 3.10 and Figure 7 necessary? 

Table 3 needs a legend or could possibly be deleted. 

Is Figure 6 necessary?

The first paragraph of section 3.11 seems out of place. Should it possibly be in the Methods section? 

Second paragraph of section 3.11 and Tables 4 and 5. The doses don't match! 25 and 50 ug vs 50 and 100 ug. 

Significant digits after the decimal point are completely crazy and range from 2-5 in Tables 4 and 5. I recommend using two for all values including the SD

Is section 3.11 really necessary since similar results are presented in figures 8 and 9. Could have a sentence or two in the methods section saying the optimal dose was determined. 

Section 3.12. It would really help the reader if a description of DT, MO, E, DR, DO, and RF was included. Or at least the meaning of these abbreviations should be included. 

Table 6 and 7. Signficant digits after the decimal point range from 0-4. I recommend one digit past the decimal point for all the values.  

Section 3.13. I am assuming that this is a repeat of section 3.11 using the optimal dose and more rabbits. If so, that should clearly be stated. Also the dose should be included in figure legends 8 and 9. 

Figure 10 and 11 legends need more detail and information. 

Numerous instances of inappropriate capitalization. Some examples on lines 19, 48, 49, 89, 103, 207, 216, 711 

Author Response

The major defect in this paper is its presentation. It is extremely difficult to follow and the paper -- especially the results -- sometimes reads like random numbers and acronyms. The authors should do a major overhaul of the paper and probably delete a lot of the modeling studies. Figure legends and table legends are generally horrendous. The authors need to slowly and carefully go through the manuscript and think about it from a potential reader's perspective and clearly describe the experiments and their results, as well as the significance of those results. 

Response: 

We are extremely sorry and as suggested, the presentation is modified thoroughly. A significant part linked to modelling is changed or deleted. Both methods and results have been rewritten keeping in view the comments of the reviewers

Figure legends and table legends are modified and clearly described the experiments, as suggested.

Comments:The abstract is too long, too technical, and too methodological. It could easily be cut to half its current size.

response: As suggested, the length of the abstract is reduced deleting too much technical and methodological points. 

Comments:First sentence of introduction should probably be expanded to at least two sentences. 

Response: Necessary changes have been incorporated

Comments: Line 75: Three synthetic peptides () derived from BM86 of R. microplus ...

Response: Necessary changes have been done

Comments: Line 84: delete comprising

Response : Deleted

Comments: Line 91: In addition is probably better the Besides

Response: Corrected

Comments : Line 112: delete Two different servers, 

Response: Deleted

Comments: Line 116: I would move the names of the two servers from line 117 to replace 'a specific server' in line 116

Response: As suggested, corrected

Comments: line 132: first use of MEP,but defined on line 180. There are numerous cases of non-standard abbreviations throughout the manuscript that are not defined

Response: The reviewer is rightly pointed out the mistakes and thus all the abbreviations have been defined

Comments: line 167: delete, if any

Response : Deleted

Comments : line 203: either delete the sentence or move it to the next section

Response : deleted

Comments: line 332: change are having to have

Response; Corrected

Comments: line 334: plasma membrane is probably better than cell membrane

Response; corrected

Comments: Table 2 needs a legend (what is P and N?

Response: Described after table

Comments: line 354: 43-100%

Response: corrected

Comments: line 373: what are the parasite proteins?

Response: Vaxijen 2.0- Server can predict antigenicity of the unknown protein on the bases of Viral, bacterial or parasite protein database as a whole

Comments: Figure 1: legend needs more information including an indication of which specific epitopes from Table 2 were used

Response: Changes are incorporated as per suggestion both in the Figure 1 and Table 2

Comments: Figure 3 has some serious formating issues, and the legend is incomprehensible. Furthermore, is the figure really necessary? 

Response: 

Figure 3 have presented in modified version.

Figure 3. was required to understand the structural chemistry of the peptides, Moreover, exposed and unexposed parts of the construct was studied for better characterization.

Comments: line 418: What is model 1?

Response: First model on the server (GalaxyRefine) among the five refined models designated as model 1

Comments: Section 3.8. What is the point of this section? What is the signifcance? Is Figure 4 necessary? What is Figure 4 showing and how does it related to VT1 and VT2 as potential vaccines? 

Response: 

The ProSA-web program (Protein Structure Analysis) is an established tool which has a large user base and is frequently employed in the refinement and validation of experimental protein structures and structure prediction and modeling. 

For designing any synthetic MEP structure prediction and validation are the foremost requirement and thus has been done.

But still reviewer thinks it is to be deleted, necessary action can be taken 

Comments: 

line 447: What is RMSD and how does it relate to the immunogenicity of VT1 and VT2.

 Is section 3.10 and Figure 7 necessary? 

Response: 

The distance between atoms is measured by the Root Mean Square Deviation (RMSD) score. A lower RMSD value suggests better stability and usually, an RMSD score ranging between 0 and 1.2 is acceptable. The stability is important for better immunogenicity of synthetic peptide

Section 3.10 has been merged with previous section, Molecular docking.

Molecular dynamics simulation indicates the stability of the vaccine candidates and the root-mean-square fluctuation (RMSF) of amino acid residues were analyzed to investigate the stability of the receptor-ligand interaction.

Figure 7. To study the comparative stability of the constructs, figure 7 was required.

But still reviewer thinks it is to be deleted, necessary action can be taken.

Comments: Table 3 needs a legend or could be deleted. 

Response : deleted

Comments: Is Figure 6 necessary?

response: 

It was used to evaluate the stability of the vaccine-receptor complex. This server run is based on Normal Mode-Analysis.

Deformability is the ability of a molecule to deform each of its residues. The B-factors in PDB files are used to measure mobility in macromolecules, including proteins. The figure 6  is also indicating stability in vaccine-TLR complex and all these data are required before synthesizing the MEP, please.

Comments: The first paragraph of section 3.11 seems out of place. Should it possibly be in the Methods section? 

Response: Changes are done as per the recommendation

Comments: Second paragraph of section 3.11 and Tables 4 and 5. The doses don't match! 25 and 50 ug vs 50 and 100 ug

response: A major mistake and we are thankful to the reviewer for pointing out the same. Necessary changes have been incorporated throughout the ms. 

Comments: Significant digits after the decimal point are completely crazy and range from 2-5 in Tables 4 and 5. I recommend using two for all values including the SD

Response: As suggested, necessary corrections have been incorporated.

Comments: Is section 3.11 really necessary since similar results are presented in figures 8 and 9. Could have a sentence or two in the methods section saying the optimal dose was determined. 

Response:

The section 3.11 is the dose standardization experiment based on IgG and IgM responses up to 42 DPI by using 2 rabbits in each dose group. Whereas, the post challenge and immune responses was monitored up to 112 DPI using 6 animals /  immunized group.

But still reviewer thinks it is to be deleted, necessary action can be taken.

Comments: Section 3.12. It would really help the reader if a description of DT, MO, E, DR, DO, and RF was included. Or at least the meaning of these abbreviations should be included. 

Response: As suggested, all the abbreviations have been expanded.

Comments:Table 6 and 7. Signficant digits after the decimal point range from 0-4. I recommend one digit past the decimal point for all the values.  

Response: Changes are done as per the recommendation

Comments: Section 3.13. I am assuming that this is a repeat of section 3.11 using the optimal dose and more rabbits. If so, that should clearly be stated. Also, the dose should be included in figure legends 8 and 9. 

Response: Section 3.13 is the Humoral immune responses to VT1 and VT2 antigens post immunization and challenge of rabbits whereas, section 3.11 is the dose standardization experiment on the basis of IgG and IgM responses up to 42 DPI

Comments: Figure 10 and 11 legends need more detail and information. 

Response : Figure legends have been modified as suggested.

Comments: Numerous instances of inappropriate capitalization. Some examples on lines 19, 48, 49, 89, 103, 207, 216, 711 

Response: We are extremely sorry for these silly mistakes. All the corrections have been incorporated

Round 2

Reviewer 4 Report

The revised version has been improved somewhat. However, it is still very difficult to read and to understand. There are still numerous cases of undefined abbreviations or defining abbreviations subsequent to their first use. Likewise, there is still a lot of inappropriate capitalizations. This could imply that the authors are rather sloppy and lazy, and certainly do not have a lot of pride. I personally would be embarrassed to have my name on such a poorly written paper. 

A structural problem with the paper is that it is rather long and combines modeling studies with experimental immunization. Presumably, the modelling studies were used to develop the vaccines. But this connection is not clearly made. I recommend to completely restructure the paper and instead of having separate Results and Discussion sections, have a single combined Results and Discussion section. This would allow a better explanation of the rationale of the modeling studies and how they lead to the development of the vaccines. The alternative would be to split the paper into two separate papers. One on modeling and vaccine development and the other on the efficacy and potential of the vaccines. 

Some (but not all) specific comments:

lines 18-19, change i.e., to called and end the sentence at VT2. Delete the rest of the sentence on line 19. 

line 26, combine the two paragraphs of the abstract

lines 50-51, rearrange the sentence to move the first phrase (to minimize the use of acaricides) to the end of the sentence

line 65, multi-epitope

line 205, change by to with

lines 218-237, it is very difficult to follow these calculations. All of the acronyms need to be defined and describes. Perhaps a table will be helpful. Is there a difference between rejection and reduction? There needs to be some explanation of efficacy (E) and why those calculations are indeed a measure of efficacy. 

line 240, inoculations

line 279, change was to were

line 309, Nine-mers and delete length

Table 2, Yes or No may be better than P or N

lines 332-333, what does the last sentence mean?

sections 3.5 and 3.6, moving the related information from the Discussion to here would be very helpful

Figure 3 and Figure 5, what exactly should the reader be seeing here. Needs more description and why this is important. Otherwise, delete. 

lines 438-441, move to the Methods section

Table 3 and Table 5 (line 453), What happened to Table 4? Delete Kinetics of comparative from both titles. Only keep two digits after decimal point

line 459, rabbits

line 462, delete an

line 463, change beginning of sentence to Nevertheless, nymphs dropped ..... What does dropped mean? And why is it important? 

throughout page 13 and elsewhere:  not necessary to say 'control group of rabbits' and 'immunized group of rabbits'. Simply say control rabbits or immunized rabbits. 

delete Mean from Table 5 and Table 6 titles

Section 3.8, the use of DT, DR, etc is difficult to follow. Especially since these acronymns have never been described very well. I recommend using descriptive terms in this section rather than the abbreviations. The legends to tables 5 and 6 should also have these descripitons of DT, DO, etc

lines 496-497, either delete of or the () enclosing the OD values

Figure 7 and 8, A and B labels not necessary since figures are clearly marked. There is a font difference in the labeling of figure 8

a lot of the uses of the word 'significant' can be deleted. It is also not necessary or helpful to include the P-values in the text

line 537-538, delete 'in ug/ml or pg/ml'

page 16, there is some out of place figure at the top of the page

Figure 10, two of panels are missing. Looks like the orphan figure at the top of the page should be in one of the panels

some examples of inappropriate capitalization: lines 14, 41, 81, 113, 329, 611

some examples of abbreviation failures: lines 81, 88, 92, 110, 145,149, 218-237

Author Response

Comment(s)

Reply

The revised version has been improved somewhat. However, it is still very difficult to read and to understand. There are still numerous cases of undefined abbreviations or defining abbreviations subsequent to their first use. Likewise, there is still a lot of inappropriate capitalizations.

We are extremely sorry, and as suggested, the presentation has been thoroughly modified. A significant part linked to the modelling study has been changed.

We are extremely sorry for this mistake after the first revision. All the abbreviations have been defined, and all inappropriate capitalizations have been modified as suggested.

lines 18-19, change i.e., to called and end the sentence at VT2. Delete the rest of the sentence on line 19. 

Necessary changes have been incorporated

line 26, combine the two paragraphs of the abstract

As suggested, combined.

lines 50-51, rearrange the sentence to move the first phrase (to minimize the use of acaricides) to the end of the sentence

Corrected

line 65, multi-epitope

Corrected

line 205, change by to with

Corrected

lines 218-237, it is very difficult to follow these calculations. All of the acronyms need to be defined and describes. Perhaps a table will be helpful. Is there a difference between rejection and reduction? There needs to be some explanation of efficacy (E) and why those calculations are indeed a measure of efficacy. 

All of the acronyms have been well defined and describes in tabulated format as suggested.

For the calculation of efficacy these parameters are  internationally accepted and have been published n number of times. Efficacy is measured on the basis of direct mortality and population limiting properties of the immunogens and accordingly these parameters were taken into consideration. . 

line 240, inoculations

Corrected

line 279, change was to were

Corrected

line 309, Nine-mers and delete length

Changes incorporated

Table 2, Yes or No may be better than P or N

Necessary changes have been incorporated

lines 332-333, what does the last sentence mean?

“Predicted pI values of VT1 and VT2 were estimated as 4.97 and 4.61, respectively, representing acidic in nature”--- to design a good peptide vaccine,  prediction of PI value of the protein is required, please.

sections 3.5 and 3.6, moving the related information from the Discussion to here would be very helpful

It has been modified to some extent.

Figure 3 and Figure 5, what exactly should the reader be seeing here. Needs more description and why this is important. Otherwise, delete. 

A major problem in structural biology is the recognition of errors in experimental and theoretical models of protein structures. The ProSA program (Protein Structure Analysis) is an established tool which has a large user base and is frequently employed in the refinement and validation of experimental protein structures and in structure prediction and modeling. Figure 3 indicates the Z-score of the refined models as - 2.23 and -4.08 which are lying inside the score range.

The VT1 and VT2 constructs with TLR-9 produced a reliably stable simulation, as shown by normal mode analysis (NMA). The simulation study was conducted to specify the motions of molecules and atoms in the construct. Figure 5A indicates the main-chain deformability simulation; the hinges are regions with high deformability for both VT1 and VT2.

lines 438-441, move to the Methods section

We think this portion is ok in the present section.

Table 3 and Table 5 (line 453), What happened to Table 4? Delete Kinetics of comparative from both titles. Only keep two digits after decimal point

We are extremely sorry for typing error in the table number (Table 4).

All changes regarding table number have been corrected.

line 459, rabbits

Corrected

line 462, delete an

line 463, change beginning of sentence to Nevertheless, nymphs dropped .....

What does dropped mean? And why is it important? 

Deleted

Line 463 have been corrected.

 In the field condition Hyalomma anatolicum is completing its life cycle in three hosts and in each stage engorged ticks are dropped on the ground after feeding and after moulting reattach to next suitable host. To assess the efficacy, number of each stage dropped from the immunized host is compared with control host. In the present experiment, rabbits were used as a model (we established the model for H. anatolicum rearing) host and post immunization the number of ticks stage (nymphs/adults) dropped from immunized and control hosts was compared. This is an internationally standardized methodology and published many times.

throughout page 13 and elsewhere:  not necessary to say 'control group of rabbits' and 'immunized group of rabbits'. Simply say control rabbits or immunized rabbits. 

Corrections have been incorporated.

delete Mean from Table 5 and Table 6 titles

Corrected

Section 3.8, the use of DT, DR, etc is difficult to follow. Especially since these acronyms have never been described very well. I recommend using descriptive terms in this section rather than the abbreviations. The legends to tables 5 and 6 should also have these descriptions of DT, DO, etc

The acronymphs were described in material and methods. All the acronyms have been described in the manuscript as per the previous suggestion (Table 2).

As suggested, description of DT, DR etc is given in Table 5 and 6

lines 496-497, either delete of or the () enclosing the OD values

Modified as suggested

Figure 7 and 8, A and B labels not necessary since figures are clearly marked.

Modified as suggested

a lot of the uses of the word 'significant' can be deleted. It is also not necessary or helpful to include the P-values in the text

Necessary changes have been incorporated

line 537-538, delete 'in ug/ml or pg/ml'

Modified as suggested

page 16, there is some out of place figure at the top of the page

Necessary changes have been incorporated

Figure 10, two of panels are missing. Looks like the orphan figure at the top of the page should be in one of the panels

Figure 10 is kept in one panel.

some examples of inappropriate capitalization: lines 14, 41, 81, 113, 329, 611

All inappropriate capitalizations have been modified as suggested.

some examples of abbreviation failures: lines 81, 88, 92, 110, 145,149, 218-237

All the abbreviations have been defined

Round 3

Reviewer 4 Report

It is still very difficult reading. But the authors seem content. Accept.